# Effects of pentoxifylline in patients with chronic Chagas cardiomyopathy: A randomized, double-blind, controlled pilot trial

**Káryta Suely Macêdo Martins**, Denise Mayumi Tanaka, Camila Godoy Fabricio, Antônio Carlos Leite de Barros Filho, Henrique Turin Moreira, Paulo Louzada Júnior, José A. Marin-Neto, Marcus Vinícius Simões*

Medical School of Ribeirao Preto – University of São Paulo, Sao Paulo, Brazil

* msimoes@fmrp.usp.br

## Abstract

### Background

Chronic Chagas cardiomyopathy (CCC) is a major public health issue in endemic areas of Latin America, representing one of the leading causes of heart failure and sudden death. The hallmark histopathological lesion of CCC is low-intensity, persistent myocarditis associated with cytokine production. Long-term use of pentoxifylline (PTX) may serve as an effective pharmacological intervention for immunomodulation, reducing inflammation and, consequently, diminishing myocardial perfusion abnormalities and thus preserving left ventricular systolic function.

### Methods

We investigated 38 patients with CCC, randomly assigned to PTX (n = 19), 400 mg 3 times a day for 6 months, or placebo (PLC) (n = 19). At baseline and post-treatment, patients underwent cytokine measurements, quality of life assessment, 2D echocardiography, and myocardial perfusion scintigraphy. After treatment, TNF-α levels in the PTX group decreased from 10.14 ± 5.5 to 8.32 ± 3.6 and from 9.12 ± 4.4 to 10.32 ± 8.5 in the PLC group (p = 0.06). Additionally, IL-10 levels increased from 2.74 ± 0.7 to 5.61 ± 8.6 in the PTX group, while in the placebo group, they decreased from 6.96 ± 11.8 to 5.50 ± 8.3 (p = 0.09); neither of these findings reached statistical significance. Also, no significant changes were observed in the echocardiographic variables after treatment. LVEF showed a modest change from 46.2% ± 7.9 to 47.4% ± 7.0 in the PTX group and from 48.2% ± 6.6 to 48.0% ± 6.9 in the PLC group (p = 0.37). No significant positive effects on myocardial perfusion were noted. However, the quality-of-life assessment documented a significant improvement of functional capacity in the PTX group.

**Data availability statement:** Aggregated summary data supporting the findings of this study are available at BioStudies (Accession code: S-BSST2186, DOI: 10.6019/S-BSST2186, https://www.ebi.ac.uk/biostudies/studies/S-BSST2186).

**Funding:** This study was funded by the Fundação de Amparo à Pesquisa do Estado de São Paulo (FAPESP) (https://fapesp.br), grant number 2016/25403-9, awarded to JAMN. The funder had no role in study design, data collection and analysis, decision to publish, or preparation of the manuscript.

**Competing interests:** The authors have declared that no competing interests exist.

## Conclusions

The results of this study suggest a potential positive effect of PTX in modulating the inflammatory profile of CCC patients. However, use of pentoxifylline in these patients did not attenuate the degree of ventricular dysfunction or reduce myocardial perfusion defects.

## Author summary

Chagas disease remains a neglected tropical disease, despite affecting millions of people. Among its clinical forms, chronic Chagas cardiomyopathy (CCC) is the most frequent and severe. This condition progresses slowly and often results in heart failure, leading to high rates of illness and death. Despite decades of research, significant gaps remain in our understanding of the mechanisms driving disease progression and the development of effective treatments. Currently, there are no therapies specifically designed for CCC; patients are treated with drugs commonly used for other heart diseases, which may not adequately address the unique features of Chagas-related heart damage. We have been working to understand how inflammation and abnormalities in myocardial perfusion contribute to the progressive loss of heart function in CCC. We conducted a clinical study to evaluate whether a drug with immune-modulating properties could offer clinical benefits. Over a six-month treatment, we observed improvements in inflammatory markers and patients' reported quality of life. However, we didn't observe significant changes in left ventricular function or myocardial perfusion. The improvements in inflammatory profile and quality of life suggest a potential role for immunomodulation in managing CCC. These findings highlight the complexity of CCC and emphasize the need for therapeutic approaches. Continued research into targeted treatments is essential to improve outcomes for patients with this neglected and serious condition.

## Introduction

Chagas disease is a parasitic disease caused by the protozoan *Trypanosoma cruzi*. Recognized as a neglected tropical disease, it is endemic in 21 Latin American countries. Due to globalization and migration patterns, it has also emerged as a growing concern in non-endemic countries [1].

It is estimated that 6–7 million people are infected worldwide, with approximately 70 million individuals living in areas of potential exposure. The average annual incidence is 30,000 new cases, with around 12,000 deaths per year [1].

Chagas disease has two phases: an acute phase, which is short in duration and usually benign, with cardiovascular manifestations being mild and uncommon, and a later chronic phase. The chronic phase, which is symptomatic and presents with

evidence of cardiac and/or digestive damage, affects approximately 30% of individuals who are chronically infected throughout their lifetime [2,3].

Cardiomyopathy is the most significant and severe manifestation of chronic Chagas disease. One of the most intriguing aspects of the natural history of Chagas disease is the long, clinically silent period, which lasts about 2–3 decades, between the initial infection and the onset of the symptomatic chronic phase [4].

Evidence from both animal and human studies indicates that two main pathogenic mechanisms contribute to the development of chronic Chagas cardiomyopathy (CCC): persistent, low-grade but virtually incessant inflammation directly elicited by tissue parasitism and and also, although with less evidence basis, exaggerated inflammatory injury probably due to autoimmune mechanisms.[5].

In addition to these central pathogenic mechanisms, clinical, necroscopic, and experimental studies have highlighted the role of coronary microvascular dysfunction in the development of chronic myocardial dysfunction in CCC. These studies also indicate a close relationship between myocardial perfusion abnormalities and inflammation, including recent studies with experimental models using high-resolution imaging [5–7]. Therefore, investigating the effect of immunomodulatory drugs on perfusion disturbances and myocardial damage in CCC is of great interest.

Several independent studies suggest that the diffuse myocarditis characterized by myocytolysis and reparative fibrosis, which is typical of chronic Chagas cardiomyopathy (CCC), exhibits feature of a delayed hypersensitivity reaction [8,9]. This reaction is marked by focal inflammatory infiltrates predominantly composed of mononuclear cells [10]. In the later stages of the disease, the destruction of cardiac muscle fibers becomes more pronounced, accompanied by extensive reparative fibrosis, leading to significant alterations in myocardial function. This process ultimately results in marked dilation of the heart chambers, manifesting clinically as dilated cardiomyopathy [11]. These inflammatory infiltrates are predominantly made up of mononuclear cells [12–14], with CD8 + T cells being the dominant type [15], producing cytokines such as IFN-γ and TNF-α [16]. In this context, the increased production of inflammatory cytokines like TNF-α, play a central role in the immunological mechanisms driving chronic myocarditis in CCC [17].

Pentoxifylline (PTX), a methylxanthine derivative, acts as a non-specific phosphodiesterase inhibitor and modulates TNF-α production by inflammatory cells [18]. Previous studies in animal models have demonstrated beneficial effects of PTX, including the reduction of peripheral edema [19] and improvement of cardiac function in Chagas disease [20], with reduced TNF-α synthesis by monocytes and macrophages [21].

Also, clinical studies testing the prolonged use of PTX in patients with non-Chagas dilated cardiomyopathy have shown favorable results, suggesting that chronic myocardial inflammation, associated with elevated serum and myocardial levels of pro-inflammatory cytokines, plays an important role in the development of chronic cardiomyopathies, regardless of etiology [22–26].

Thus, the present study aims to evaluate the use of PTX as an effective pharmacological intervention to positively impact the reduction of myocardial perfusion abnormalities and improve regional and global left ventricular systolic function.

## Methods

### Ethics statement

This study was reviewed and approved by the Ethics Committee on Human Research of the Hospital das Clínicas de Ribeirão Preto and the Ribeirão Preto Medical School, University of São Paulo (approval number: 2.281.063). All participants provided formal written informed consent prior to their inclusion in the study, in accordance with the Declaration of Helsinki.

## Study population

Patients were recruited from the specialized outpatient clinics at the Cardiology Center of the Hospital das Clínicas, Faculty of Medicine of Ribeirão Preto. The study included patients with a confirmed diagnosis of chronic Chagas cardiomyopathy (CCC), based on: a) positive epidemiology and at least two positive serological tests (indirect immunofluorescence or enzyme-linked immunosorbent assay – ELISA); b) typical segmental wall motion abnormalities of the left ventricle, as evidenced by transthoracic echocardiography, characterizing chronic myocardial involvement due to the disease. Participants had a preserved or mildly reduced left ventricular ejection fraction (LVEF ≥ 35%) and could present heart failure symptoms, corresponding to New York Heart Association (NYHA) Functional Class I or II.

Patients were excluded if they had another etiology for myocardial dysfunction, such as alcoholism, prior myocardial infarction, known coronary artery disease, use of cardiotoxic drugs or illicit substances, peripartum cardiomyopathy, primary valvular heart disease, or pericardial diseases. Additionally, individuals with comorbidities that compromised functional capacity, such as severe chronic obstructive pulmonary disease (COPD), severe liver dysfunction collagen diseases, or untreated thyroid dysfunction, were also excluded.

This clinical trial is registered in the Brazilian Registry of Clinical Trials (ReBEC), ID: RBR-8k345j and it can be verified at the following address: https://ensaiosclinicos.gov.br/rg/RBR-8k345j

## Study procedures and design

After obtaining informed consent, patients were randomized in a 1:1 ratio to the intervention and control groups using a computer-generated algorithm for simple randomization. Randomization was conducted by an independent researcher who was not involved in participant enrollment or study procedures, ensuring allocation concealment.

The investigational groups were: 1) PTX group, receiving pentoxifylline 400 mg every 8 hours for 6 months; and 2) Placebo group, receiving a placebo tablet identical in appearance every 8 hours for 6 months. Both patients and investigators directly involved in patient care and assessments were blinded to the treatment group allocation throughout the study development.

Patients included in the study underwent baseline evaluation, which included clinical assessment, transthoracic echocardiography, myocardial perfusion scintigraphy with Sestamibi labeled with Technetium-99m (99mTc-Sestamibi) at rest and stress, and serum cytokine level measurements: IFN-γ, IL-10, IL-6, TNF-α, TGF-β, and endothelin-1. After 6 months of treatment, all patients underwent final evaluation using the same methods as the baseline assessment.

### Specific evaluation methods

**Serum cytokine measurements.** Peripheral blood samples were collected from all study participants before and after treatment with pentoxifylline or placebo to measure cytokine levels of IFN-γ, IL-10, IL-6, TNF-α, TGF-β, and endothelin-1. Plasma samples were processed and stored at -80°C until analysis. Analyses were conducted at the Molecular Immunology Laboratory of the Hospital das Clínicas of Ribeirão Preto – FMRP/USP. Cytokines and inflammatory markers (TNF-α, IFN-γ, IL-6, IL-10, TGF-β, and endothelin-1) were quantified in plasma (25 µL) using the Milliplex MAP Human Cytokine/Chemokine magnetic bead panel (#HAGP1MAG-12K; #HCYTOMAG-60K; #TGFBMAG-64K, EDM Millipore, Billerica, MA, USA). The assay was performed in 96-well plates according to the manufacturer's instructions, and the results were expressed in pg/mL. Each assay plate comprised 7 standards, 2 positive controls, 2 blank wells, and 76 samples. Results were analyzed using the Luminex-200 System (Luminex, Austin, TX, USA) and reported using xPOTENT software version 3.1. Cytokine concentrations were calculated using a five-parameter logistic curve-fitting method for TNF-α, IFN-γ, IL-6, and IL-10, and a cubic spline method for TGF-β, employing the median fluorescence intensity (MFI) technique. All data were corrected using Milliplex Analyst software [27].

**Cardiac biomarkers.** Additionally, serum NT-proBNP levels were measured in all study participants before and after treatment.

**Echocardiography.** Echocardiographic examinations were performed using GE ultrasound equipment, models Vivid E9 or S6, with sector transducers capable of harmonic imaging acquisition. Images were analyzed offline (EchoPac) using dedicated software capable of analyzing not only longitudinal deformation but also circumferential, radial, and torsional deformation.

Conventional two-dimensional echocardiography was performed at rest with the patient in the left lateral decubitus position, acquiring conventional projections, including those dedicated to the analysis of the right ventricle (RV) and short-axis views of the left ventricle (LV) at basal, mid-ventricular, and apical levels. Image acquisition was guided by electrocardiographic tracings recorded on the equipment. Image archiving was performed in both DICOM format and GE-specific RAW data format.

Several echocardiographic parameters were measured to assess: cardiac chamber dimensions (left ventricular end-diastolic diameter (LVEDD), left ventricular end-diastolic volume (LVEDV), and left atrial volume (LAV)), global systolic function of the LV (left ventricular ejection fraction (LVEF)), regional LV function (segmental wall motion (SWM)), right ventricular function (TAPSE), diastolic dysfunction and filling pressures (E/e'), and ventricular remodeling (posterior wall thickness, septal thickness, and left ventricular mass index).

Segmental wall motion (SWMI) was calculated using the 17-segment model [28]. A parasternal long-axis projection and three short-axis cuts (basal, mid-ventricular, and apical) were used for segmentation. SWM was visually graded for each segment and assigned scores: 1 for normal, 2 for hypokinetic, 3 for akinetic, and 4 for dyskinetic. The SWM score was calculated by summing the scores assigned to each segment and dividing by the total number of segments evaluated.

Echocardiographic exams were performed by an investigator blinded to the treatment group and study phase.

**Myocardial perfusion.** Scintigraphy tomographic images (SPECT) were acquired following the injection of Tc-99m sestamibi radiopharmaceutical at rest (10 mCi) and during pharmacological stress using adenosine (30 mCi), following universally established and previously published protocols [29]. Rest and stress images were acquired 1 hour after the radiopharmaceutical injection. Imaging was performed at the Nuclear Medicine Section of the HC-FMRP-USP using a BrightView gamma camera (Philips, Netherlands). Images were reconstructed in the three orthogonal planes using an iterative reconstruction method (OSEM) and analyzed by constructing polar maps with MunichHeart software, enabling semi-automated calculation of perfusion defect areas in stress and rest images. Perfusion defects were defined as areas with pixel uptake below 2.5 standard deviations compared to normal uptake in a database of healthy individuals. This automatic analysis was performed using Polar Map Calculator software, part of the MunichHeart processing package. The tool enabled the calculation of resting perfusion defect areas, stress defect areas, and the difference between stress and rest, constituting the area of reversible ischemic perfusion defects.

Perfusion study analyses were also performed by an investigator blinded to the group and study phase to which the patients belonged. Figs 1 and 2 illustrate the image processing performed by the software.

**Eletrocardiogram.** Twelve-lead electrocardiograms were obtained in a hospital setting using the standard technique. The following parameters were assessed: heart rhythm, atrioventricular and intraventricular conduction disturbances, the presence of ventricular and supraventricular arrhythmias, QRS voltage, the presence of electrically inactive zones, corrected QT interval, PR interval, and QRS dispersion.

**General health assessment.** The SF-36 is a validated instrument in Brazil, easy to understand, and has been used in patients with various diseases [30]. It contains 36 questions: 10 regarding physical aspects, 2 about social aspects, 4 related to limitations due to physical health problems, 3 related to limitations due to emotional problems, 5 concerning mental health, 4 on vitality (energy/fatigue), 2 on pain, 5 on general health perceptions, and 1 on changes in health status. The score ranges from zero to 100, with higher scores indicating better health status. The eight domains assessed

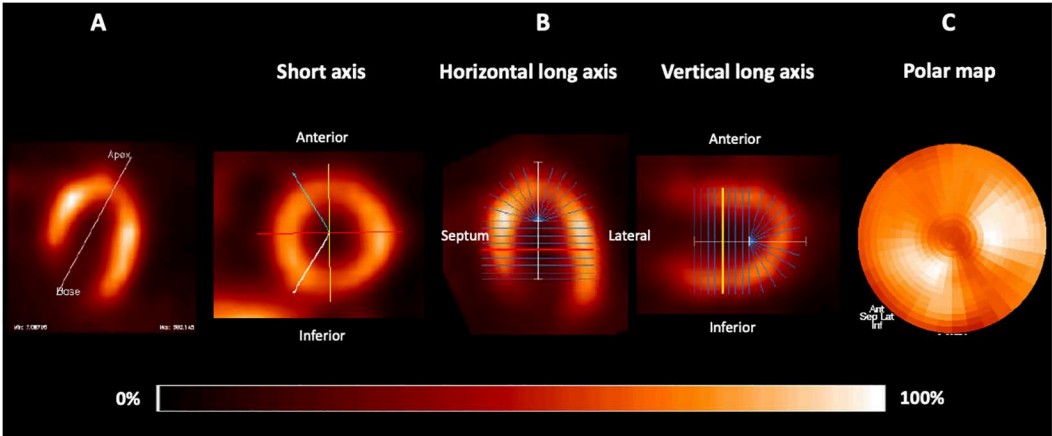

**Fig 1. Illustrative images of the processing of a myocardial perfusion study in a patient with Chagas disease.** A: Representation of the alignment in orthogonal planes of the tomographic slice orientations of the left ventricle. B: Definition and identification of left ventricular regions for the construction of the polar map. C: Resulting polar map.

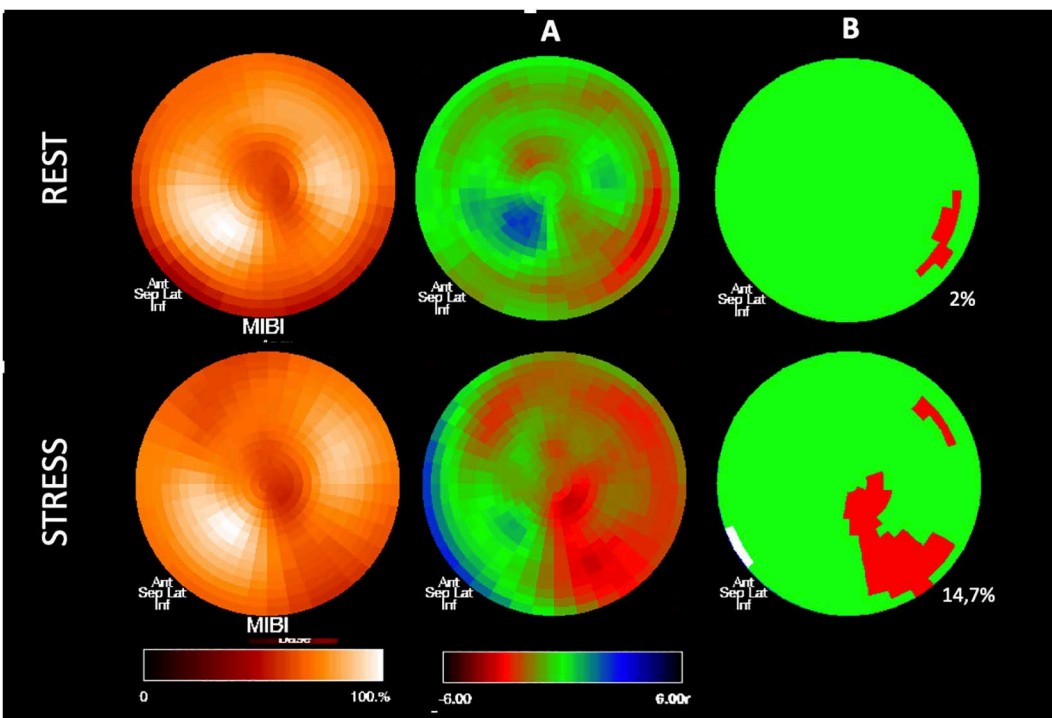

**Fig 2. Illustrative images of the quantitative processing of polar maps from a myocardial perfusion study in one of the investigated patients.** A: Comparison of radiotracer uptake areas with a database of healthy individuals at rest and during stress, respectively, showing pixel distribution according to standard deviations from the mean value observed in each pixel of the normal database. B: Identification of regions with uptake thresholds below 25 standard deviations in rest and stress images, identifying significant perfusion defect areas compared to the normal database (areas marked in red). The extent of reversible ischemic perfusion defect is calculated by subtracting the rest defect area from the stress defect area (reversible defect area = stress defect area − rest defect area).

are: functional capacity, physical aspects, pain, general health, vitality, social aspects, emotional aspects, and mental health. The questionnaires were administered by a blinded evaluator to all individuals included.

## Statistical analysis

The results for continuous variables are presented as mean ± standard deviation (SD) for Gaussian distribution variables or as median and interquartile range for non-Gaussian distribution variables. Nominal variables are expressed as absolute (n) and relative frequency (%).

The Shapiro-Wilk test was used to assess the Gaussian distribution of the variables. For comparing means between two independent samples at the same time point, the Mann-Whitney U test was applied for non-normally distributed variables, while the unpaired t-test was used for normally distributed variables. Fisher's exact test was employed to assess the heterogeneity of frequency distributions.

The effect of treatment over time in both groups was analyzed using mixed-effects ANOVA models, which accounted for the interaction between independent variables: type of treatment and time. Statistical significance was set at a two-tailed p-value < 0.05. All statistical analyses were conducted using GraphPad Prism 9.0 software.

We conducted a principal component analysis (PCA) for a multiparametric assessment of treatment effects across six standardized change parameters: LVEF, reversible myocardial perfusion defects, IL-10, TNF-alfa, NT-proBNP, and SF-36 functional capacity. Individual component scores were then compared between the pentoxifylline and placebo arms using the student t-test or two-sample Wilcoxon rank-sum test, as appropriate.

A total sample size of 46 patients (23 patients per investigational group) was calculated to detect a 5% difference in left ventricular ejection fraction (LVEF) between the groups at the end of the treatment [31]. We assumed a standard deviation of 6% for serial LVEF measurements, with 80% power and a 95% confidence interval (sample size calculation performed using www.openepi.com).

## Results

### Patient population sample description

A total of 74 patients were initially screened for eligibility in the study, of which 46 were enrolled and randomized to one of the investigational groups. During the course of the study, 8 patients discontinued follow-up, 4 patients from each group. The reasons for dropout were varied: 1 patient experienced intolerable side effects from the medication (refractory nausea despite symptomatic treatment), 3 patients moved to other city and were unable to attend in-person follow-up visits, 2 patients just opted to withdraw from the study without any alleged reason, and 2 patients died (1 in each group). Thus, a total of 19 patients in each intervention arm completed all study procedures as planned. The patient sample flow is depicted in Fig 3, using a CONSORT diagram.

The study was conducted between October 2017 and March 2020. An expansion of the number of patients to be included, to account for unforeseen follow-up losses, was approved by the Research Ethics Committee in October 2019 through a study protocol amendment. However, the restriction imposed by the COVID-19 pandemic prevented the effective increase in the number of patients recruited.

Table 1 summarizes the clinical and demographic data of the patients included in the study. The majority of the patients were male (65.8%). The mean age was 61.0 ± 11.1 years in the placebo group and 59.7 ± 15.0 years in the pentoxifylline group (p = 0.8). The mean left ventricular ejection fraction (LVEF) was 48.4 ± 6.6% in the placebo group and 46.2 ± 8.0% in the pentoxifylline group, with no statistically significant difference between the groups (p > 0.05). The distribution of patients within different LVEF ranges also did not differ between the two groups. The most frequent comorbidity found in both groups was systemic arterial hypertension (SAH). SAH was the only comorbidity that showed a significantly different distribution between the groups (p = 0.0008), being significantly more prevalent in the placebo group. Among the

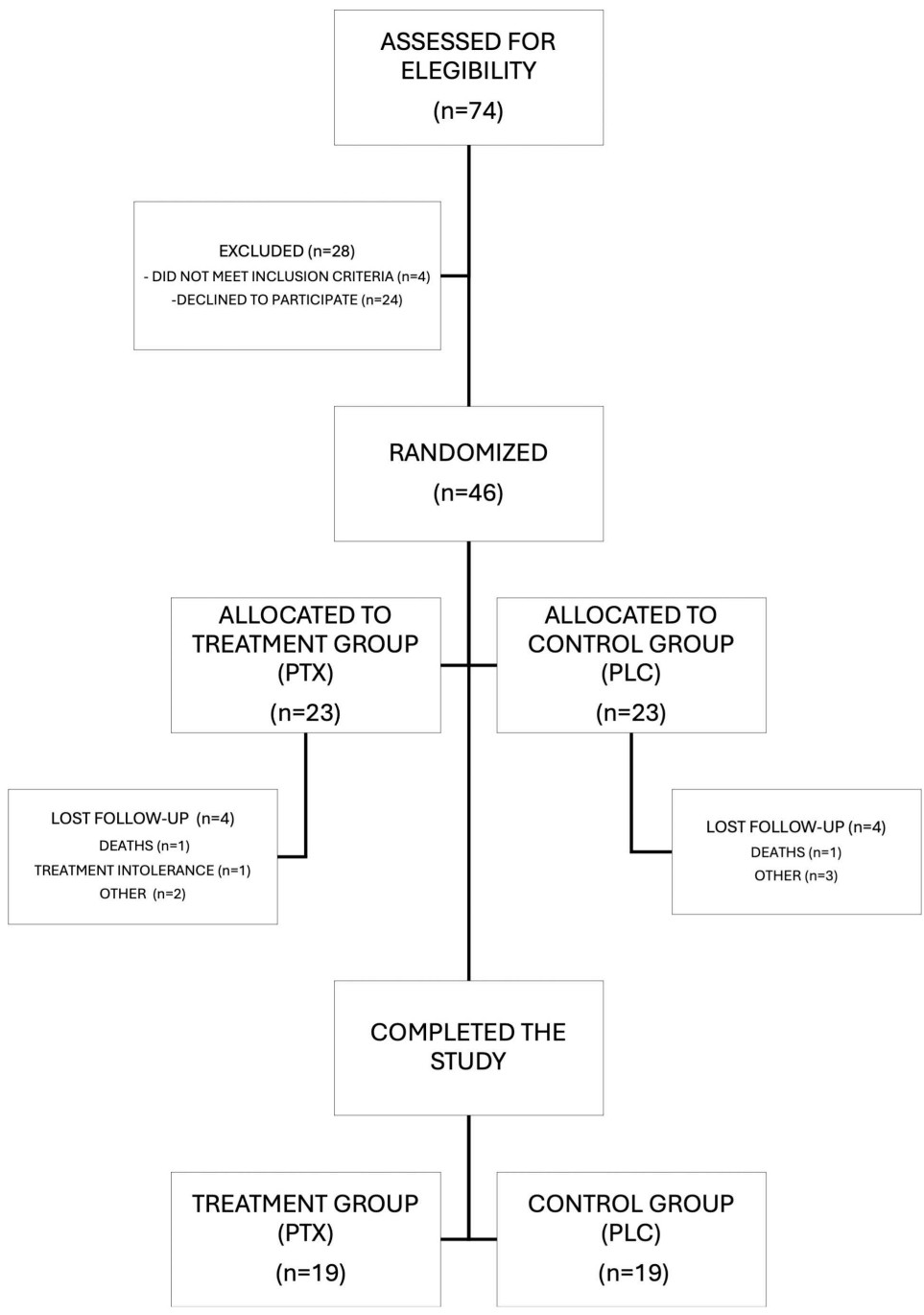

**Fig 3. CONSORT diagram illustrating patient flow during the study.**

**Table 1. Clinical and Demographic Characteristics of the Study Population.**

| Characteristic | Placebo (n=19) | Pentoxifylline (n=19) | p-value |
|---|---|---|---|
| **Male** | 10 (52.6%) | 15 (78.9%) | 0.08 |
| **Age (years)** | 61.0±11.1 | 59.7±14.8 | 0.80 |
| **White** | 10 (52.6%) | 12 (63.2%) | 0.50 |
| **BMI (kg/m²)** | 28.1±5.5 | 27.6±5.0 | 0.70 |
| **SBP (mmHg)** | 119.9±17.1 | 115.1±17.1 | 0.40 |
| **DBP (mmHg)** | 75.4±12.8 | 72.1±14.9 | 0.50 |
| **Heart Rate (bpm)** | 67.1±10.7 | 66.7±11.0 | 0.90 |
| **LVEF (%)** | 48.4±6.6 | 46.2±8.0 | 0.40 |
| >50% | 8 (42.1%) | 4 (21.0%) | 0.19 |
| 40-50% | 9 (47.4%) | 12 (63.2%) | 0.33 |
| **NYHA functional class** | | | |
| CFI | 8 (42.1%) | 12 (63.2%) | 0,33 |
| CFII | 11 (57.9%) | 7 (36.8%) | 0,33 |
| **Comorbidities** | | | |
| Hypertension | 17 (89.5%) | 7 (36.8%) | 0.0008 |
| Diabetes Mellitus | 5 (26.3%) | 1 (5.3%) | 0.07 |
| Dyslipidemia | 7 (36.8%) | 6 (31.6%) | 0.73 |
| Obesity | 5 (26.3%) | 4 (21.0%) | 0.70 |
| Stroke | 3 (15.8%) | 4 (21.0%) | 0.68 |
| Atrial Fibrillation | 3 (15.8%) | 3 (15.8%) | 1.00 |
| **Medications** | | | |
| ACE Inhibitors | 14 (73.7%) | 13 (68.4%) | 0.72 |
| ARB | 7 (36.8%) | 4 (21.0%) | 0.28 |
| Beta-Blockers | 15 (79%) | 14 (73.7%) | 0.70 |
| Aldosterone Antagonists | 2 (10.5%) | 4 (21.0%) | 0.37 |
| Diuretics | 6 (31.6%) | 6 (31.6%) | 1.00 |
| Amiodarone | 8 (42.1%) | 6 (31.6%) | 0.50 |

**BMI:** body mass index; **SBP:** systolic blood pressure; **DBP:** diastolic blood pressure; **HR:** heart rate; **LVEF:** left ventricular ejection fraction; **NYHA I:** New York Heart Association Functional Class I; **NYHA II:** New York Heart Association Functional Class II; **AF:** atrial fibrillation; **PM:** pacemaker; **SAH:** systemic arterial hypertension; **DM:** diabetes mellitus; **DLD:** dyslipidemia; **Stroke:** cerebrovascular accident; **ACEI:** angiotensin-converting enzyme inhibitors; **ARB:** angiotensin receptor blockers; **RBBB:** right bundle branch block; **LAFB:** left anterior fascicular block; **ICD:** implantable cardioverter-defibrillator.

medications used, beta-blockers were the most common, followed by angiotensin-converting enzyme inhibitors. The percentage of patients using the various medications was comparable between the two groups.

### Serum cytokine levels

At baseline, we did not observe any differences in serum cytokine levels between the studied groups. After treatment, the PTX-treated group exhibited a trend toward increased interleukin-10 levels, Fig 4A, while levels remained numerically stable in the control group. However, the comparative analysis between the groups did not reach statistical significance (p=0.2). A reduction in serum TNF-α levels was also observed following pentoxifylline treatment (10.14±5.5 vs. 8.32±3.6), whereas the placebo group showed an increase in this cytokine (9.12±4.4 vs. 10.32±8.5) (p=0.06), Fig 4B. Although the changes in cytokine levels did not reach statistical significance, the results suggest a potential association, especially considering the observed effect size (Table 2).

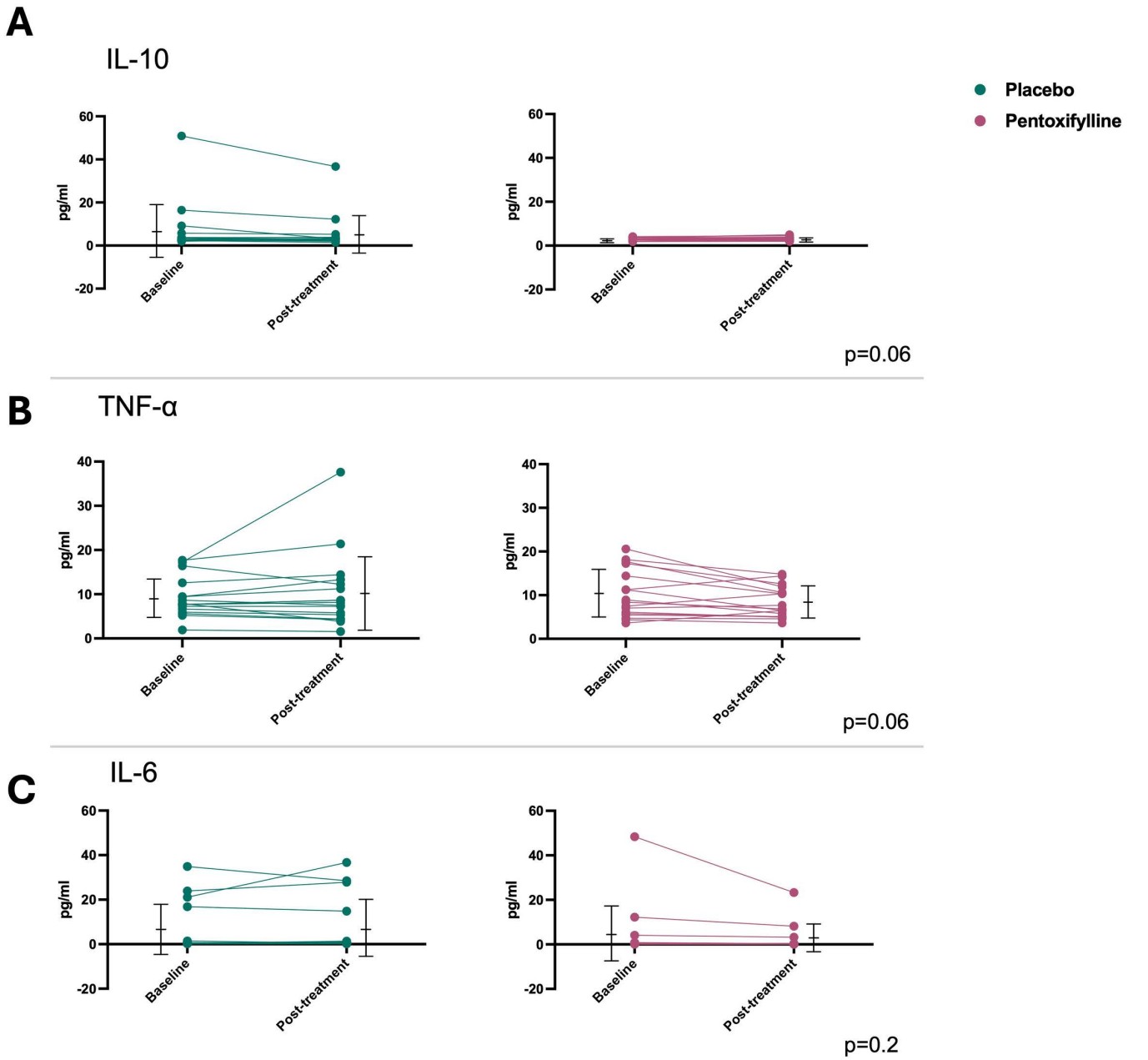

**Fig 4. Line graph showing the values of inflammatory cytokines at baseline and after treatment.** A: IL-10; B: TNF-α; C: IL-6. p=ANOVA Mixed Models.

## Cardiac biomarkers- NT pro-BNP

Analyses using mixed-model ANOVA for repeated measures did not show any significant differences in the levels of this marker. In the placebo group, values ranged from 1207.1±2343.5 at baseline to 2762.8±8098.4 at the final assessment, while in the PTX group, values ranged from 654.9±1226.9 at baseline to 508.4±464.7 at the final assessment (p=0.25),

**Table 2. Baseline and Post-Treatment Serum Cytokine Levels in Each Investigational Group.**

| Cytokine | Placebo (n = 17) Baseline | Placebo (n = 17) Post-Treatment | Pentoxifylline (n = 17) Baseline | Pentoxifylline (n = 17) Post-treatment | p value (ANOVA Mixed Models) |
|---|---|---|---|---|---|
| IFN-γ (pg/mL) | 3.05 ± 2.9 | 3.19 ± 2.8 | 2.53 ± 1.8 | 2.60 ± 3.8 | 0.93 |
| IL-10 (pg/mL) | 6.96 ± 11.8 | 5.50 ± 8.3 | 2.74 ± 0.7 | 5.61 ± 8.6 | 0.06 |
| IL-6 (pg/mL) | 6.12 ± 1.08 | 6.76 ± 1.22 | 4.07 ± 1.17 | 2.2 ± 5.8 | 0.20 |
| TNF-α (pg/mL) | 9.12 ± 4.4 | 10.32 ± 8.5 | 10.14 ± 5.5 | 8.32 ± 3.6 | 0.06 |
| ED-1 (pg/mL) | 3.00 ± 6.4 | 5.00 ± 9.1 | 4.37 ± 98 | 5.58 ± 15.2 | 0.86 |
| TGF-β (pg/mL) | 13890.6 ± 7334.5 | 12172.0 ± 6309.7 | 10240.8 ± 7863.2 | 11407.7 ± 6600.5 | 0.27 |

**IL-10:** interleukin-10; **IL-6:** interleukin-6; **TNF-alpha:** tumor necrosis factor-alpha; **ED-1:** endothelin-1; **TGF-beta:** transforming growth factor-beta.

Table 3. There was also no difference between the baseline values of the placebo and pentoxifylline groups (p = 0.914 – Sidak's multiple comparison post-test).

## Assessment of cardiac structure and function

In our cohort, no significant improvement was observed in either LVEF or left ventricular dimensions in patients treated with PTX (Table 4). Additionally, no differences were found in the variables assessing diastolic function (E/e'), right ventricular systolic function (TAPSE), or the indexed left atrial volume (LAVi). Mixed-model ANOVA did not demonstrate a significant interaction between the study groups and the time interval between baseline and final evaluations (p > 0.05) for these variables. When assessing segmental wall motion using the wall motion score index (WMSI), no remarkable changes were observed in either group after treatment.

## Myocardial perfusion assessment

At baseline, 9 of the 19 patients in the placebo group (47.3%) exhibited resting perfusion defects, with the basal anterolateral segment being the most affected (31.5%), followed by the mid-anterolateral (15.8%) and apical segments (15.8%). Regarding reversible (ischemic) defects, 13 patients in the placebo group (68.4%) exhibited this abnormality, with the mid-anterolateral segment being the most affected (37%).

In the PTX group, there was also an even higher prevalence of resting defects at baseline, with 16 of the 19 patients (84%) having at least one affected segment. The apical segment (53%) and basal anterolateral segment (42%) were the most commonly affected. Similarly, in this group, 58% of patients exhibited reversible defects, with the basal anterolateral, mid-anterolateral, apical lateral, and apical inferior segments being the most commonly affected Table 5 presents the values of the areas of myocardial perfusion defects relative to the total left ventricular area at rest and post-stress for each experimental group, at the baseline and post-treatment evaluations. The table also includes the areas of reversible myocardial perfusion defects.

No significant differences were observed between the myocardial perfusion defect values in the studied groups under any of the conditions described above, Fig 5.

## Assessment of health-related quality of life

The mean scores for the domains of the SF-36 quality of life questionnaire are presented in Table 6.

At the baseline evaluation, patients in both groups reported a good perception of functional capacity and physical aspects. However, both groups had a poor perception regarding general health and vitality. Social, emotional, and mental health aspects were usually preserved in most patients.

**Table 3. Baseline and Post-Treatment Serum Levels of NT-proBNP in Each Investigational Group.**

| | Placebo (n = 17) Baseline | Placebo (n = 17) Post-Treatment | Pentoxifylline (n = 17) Baseline | Pentoxifylline (n = 17) Post-Treatment | p-value (ANOVA Mixed Models) |
|---|---|---|---|---|---|
| **NT-proBNP** | 1207.1 ± 2343.5 | 2762.8098.4 | 654.9 ± 1226.9 | 508.4 ± 464.7 | 0.25 |

NT-proBNP: N-terminal pro-B-type natriuretic peptide.

**Table 4. Baseline and Post-Treatment Results of Echocardiographic Variables in Each Experimental Group.**

| Variable | Placebo (n = 19) Baseline | Placebo (n = 19) Post-treatment | Pentoxifyl-line (n = 19) Baseline | Pentoxifylline (n = 19) Post-treament | p-value (ANOVA Mixed Models) |
|---|---|---|---|---|---|
| **LVEF (%)** | 48.2 ± 6.6 | 48.0 ± 6.9 | 46.2 ± 7.9 | 47.4 ± 7.0 | 0.37 |
| **WMSI** | 1.6 ± 0.3 | 1.7 ± 0.4 | 1.7 ± 0.4 | 1.7 ± 0.4 | 0.10 |
| **LVEDD (mm/m²)** | 30.7 ± 5.6 | 30.7 ± 5.7 | 30.4 ± 4.4 | 30.7 ± 4.1 | 0.79 |
| **Posterior Wall (mm)** | 8.5 ± 1.7 | 8.1 ± 1.4 | 8.1 ± 1.1 | 8.2 ± 1.1 | 0.15 |
| **Septum (mm)** | 9.1 ± 2.0 | 9.3 ± 2.0 | 8.7 ± 1.4 | 8.9 ± 1.3 | 0.09 |
| **LVMI (g/m²)** | 56.4 ± 18.5 | 55.4 ± 16.7 | 51.5 ± 12.2 | 54.1 ± 11.8 | 0.13 |
| **LA Volume Index (mL/m²)** | 27.3 ± 7.7 | 30.0 ± 12.7 | 25.1 ± 13.0 | 25.3 ± 7.7 | 0.30 |
| **RV End-Diastolic Volume Index (mL/m²)** | 65.8 ± 19.0 | 76.7 ± 25.0 | 73.6 ± 19.3 | 75.4 ± 15.8 | 0.10 |
| **E/e' Ratio** | 10.2 ± 3.7 | 10.1 ± 4.5 | 9.4 ± 2.7 | 9.4 ± 2.5 | 0.30 |
| **TAPSE (mm)** | 19.9 ± 3.4 | 19.8 ± 2.9 | 19.7 ± 3.0 | 18.3 ± 1.7 | 0.20 |

**LVEF:** left ventricular ejection fraction; **WMSI:** wall motion score index; **LVEDD (indexed):** left ventricular end-diastolic diameter indexed to body surface area; **PW:** posterior wall; **LVMI (indexed):** left ventricular mass index; **LAVi:** left atrial volume indexed to body surface area; **LVEDVi:** left ventricular end-diastolic volume indexed to body surface area; **E/e':** variable assessing left ventricular diastolic function; **TAPSE:** tricuspid annular plane systolic excursion.

**Table 5. Baseline and Post-Treatment Myocardial Perfusion Defects Expressed as a Percentage of the Total Left Ventricular Surface Area in Each Experimental Group.**

| Perfusion Defect | Placebo (n = 19) Baseline | Placebo (n = 19) Post-treatment | Pentoxifylline (n = 19) Baseline | Pentoxifylline (n = 19) Post-treatment | p-value (ANOVA Mixed Models) |
|---|---|---|---|---|---|
| **Rest** | 9.0% ± 8.5% | 8.0% ± 9.7% | 16.2% ± 11.2% | 13.2% ± 10.5% | 0.32 |
| **Stress** | 14.1% ± 12.4% | 11.3% ± 11.8% | 23.2% ± 13.7% | 18.8% ± 16.5% | 0.75 |
| **Reversible** | 5.7% ± 9.5% | 5.0% ± 12.7% | 7.0% ± 10.7% | 5.6% ± 14.5% | 0.89 |

In the post-treatment evaluation, patients treated with pentoxifylline reported improvement in all domains except for mental health, which remained unchanged. In the placebo group, patients reported improvement in all aspects evaluated by the questionnaire.

Mixed-model ANOVA showed a significant interaction (p = 0.01) for the domain assessing functional capacity when comparing the baseline and final states in the PLC and PTX groups. No significant interactions were observed in the other domains.

## Electrocardiogram

The results of the baseline and post-treatment electrocardiographic analysis are presented in Tables 7 and 8. No significant differences were observed between groups regarding the presence of atrial fibrillation, pacemaker rhythm,

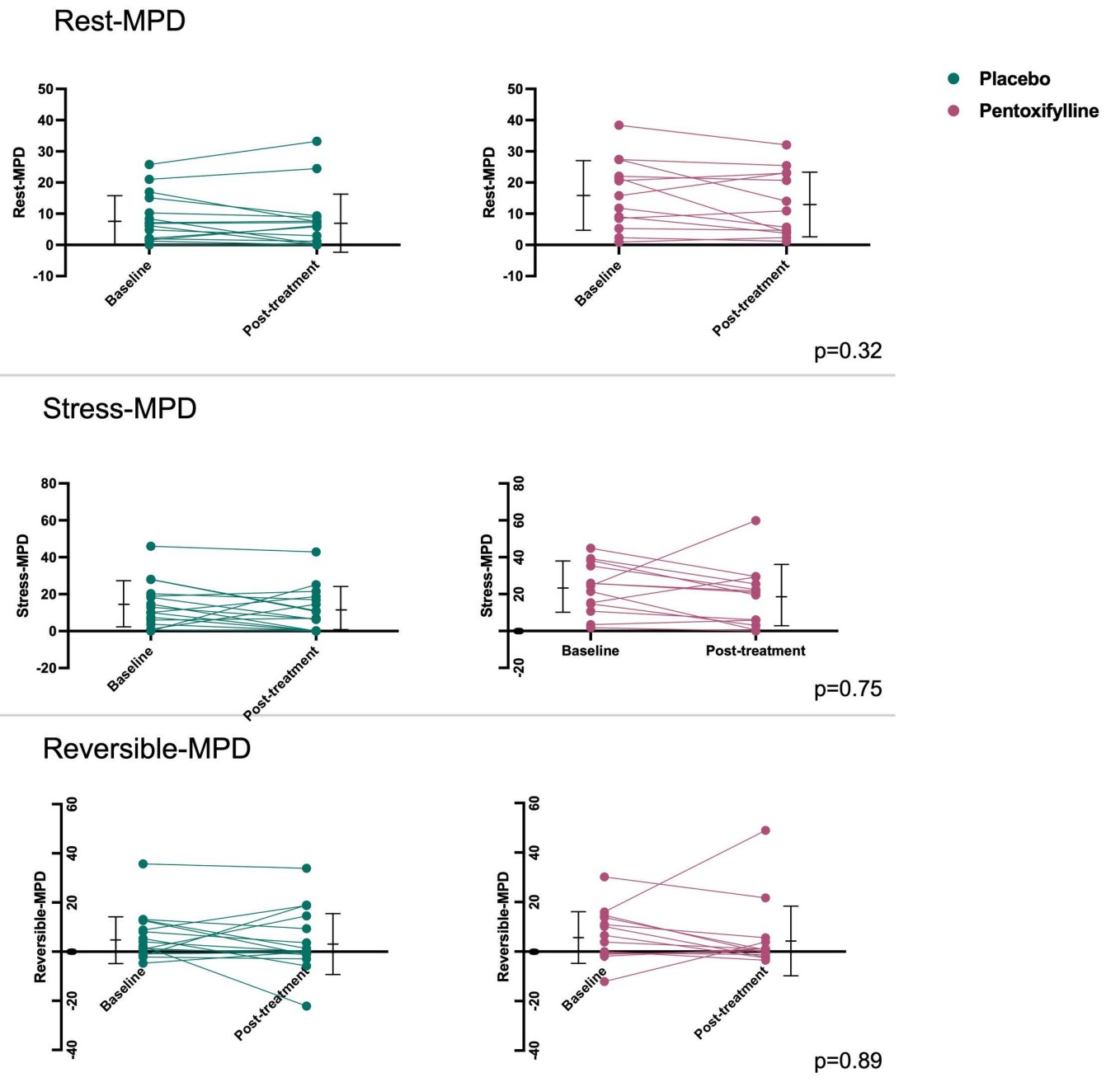

**Fig 5. Line graph showing the values of myocardial perfusion defects in the study groups. p=ANOVA Mixed Models.**

intraventricular conduction disturbances (complete right bundle branch block, complete left bundle branch block, or left anterior fascicular block), ventricular and supraventricular ectopic beats, electrically inactive zones, or low voltage. Regarding intraventricular conduction disturbances, in the placebo group, 9 patients (47.4%) presented complete right bundle branch block and 1 patient (5.3%) complete left bundle branch block. In the pentoxifylline group, 9 patients (47.4%) presented complete right bundle branch block, and none presented complete left bundle branch block. Left anterior

**Table 6. Variables obtained from the SF-36 quality of life questionnaire in each experimental group.**

| Domain | Placebo (Baseline) | Placebo (Post-treatment) | Pentoxifylline (Baseline) | Pentoxifylline (Post-treatment) | p-value |
|---|---|---|---|---|---|
| CF (%) | 73.4±16.5 | 68.2±15.8 | 70.0±23.1 | 76.7±18.1 | 0.01 |
| AF (%) | 81.6±24.2 | 85.5±23.6 | 87.5±25.6 | 96.2±8.9 | 0.6 |
| Pain (%) | 72.6±30.6 | 78.6±25.1 | 78.3±29.6 | 81.6±30.8 | 0.8 |
| GHS (%) | 47.6±20.0 | 57.2±19.9 | 50.6±22.5 | 56.8±22.4 | 0.6 |
| Vitality (%) | 57.9±21.6 | 62.2±22.5 | 63.9±24.6 | 67.7±22.9 | 0.9 |
| SA (%) | 88.8±16.1 | 87.5±22.8 | 87.5±21.9 | 95.1±10.6 | 0.3 |
| EA (%) | 83.3±25.0 | 93.0±14.0 | 86.6±24.4 | 96.3±11.2 | 0.9 |
| MH (%) | 68.9±13.7 | 71.6±20.1 | 75.6±17.1 | 76.7±14.7 | 0.7 |

Values are presented as mean±SD. CF=functional capacity; AF=physical aspect; GHS=general health status; SA=social aspect; EA=emotional aspect; MH=mental health. The p-value refers to mixed-effects analysis.

**Table 7. Baseline and post-treatment results of categorical electrocardiographic variables in each study group.**

| Variables | Placebo (n=19) | | p McNemar's test | Pentoxifyllin (n=19) | | p McNemar's test |
|---|---|---|---|---|---|---|
| | *Baseline* | *Post-treatment* | | *Baseline* | *Post-treatment* | |
| **PM** | 3 (15.8%) | 3 (15.8%) | | 5 (26.3%) | 5 (26.3%) | |
| **AF** | 1 (5.3%) | 1 (5.3%) | | 1 (5.3%) | 2 (10.5%) | 1.0 |
| **RBBB** | 9 (47.4%) | 10 (52.6%) | 1.0 | 7 (36.8%) | 7 (36.8%) | |
| **LBBB** | 0 (0%) | 0 (0%) | | 0 (0%) | 1 (5.3%) | |
| **LAFB** | 7 (36.8%) | 10 (52.6%) | 0.4 | 7 (36.8%) | 7 (36.8%) | |
| **VE** | 5 (26.3%) | 1 (5.3%) | | 4 (21.1%) | 3 (15.8%) | 1.0 |
| **SVE** | 0 (0%) | 0 (0%) | | 0 (0%) | 0 (0%) | |
| **EIZ** | 2 (10.5%) | 1 (5.3%) | 1.0 | 0 (0%) | 1 (5.3%) | |
| **Low Voltage** | 1 (5.3%) | 2 (10.5%) | 1.0 | 3 (15.8%) | 2 (10.5%) | 1.0 |

PM=Pacemaker rhythm; AF=Atrial fibrillation; RBBB=Right bundle branch block; LBBB=Left bundle branch block; LAFB=Left anterior fascicular block; VE=Ventricular ectopic beats; SVE=Supraventricular ectopic beats; EIZ=Electrically inactive zone.

**Table 8. Baseline and post-treatment results of quantitative electrocardiographic variables in each study group.**

| Variable | Placebo (n=19) | | Pentoxifylline (n=19) | | p mixed-ANOVA |
|---|---|---|---|---|---|
| | *Baseline* | *Post-treatment* | *Baseline* | *Post-treatment* | |
| **Corrected QT (ms)** | 431±51.7 | 419.3±43.6 | 413.4±45.4 | 403.3±39.4 | 0.9 |
| **QT Dispersion (ms)** | 43.1±15.4 | 35.6±12.1 | 37.9±17.6 | 36.9±16.5 | 0.4 |
| **QRSd (ms)** | 131.6±31.5 | 134.7±39.8 | 119.5±31.5 | 119.5±31.5 | 0.2 |
| **PR Interval (ms)** | 180±58.4 | 188.7±56.6 | 172.5±41.4 | 183.3±35.8 | 0.8 |

QT corrected (ms): Corrected QT interval; QT dispersion (ms): QT interval dispersion; QRS duration (ms): Duration of the QRS complex; PR interval (ms): PR interval duration. Values are expressed as mean±standard deviation. *p*: p-value for the interaction term (time×treatment) calculated using repeated measures ANOVA (mixed-model ANOVA).

fascicular block was identified in 7 patients (36.8%) in the placebo group and in 8 patients (42.1%) in the pentoxifylline group. Ventricular ectopic beats were infrequent in both groups, while supraventricular ectopic beats were not recorded in any patient. In addition, no significant effects of pentoxifylline were observed on the corrected QT interval, QT dispersion, QRS duration, or PR interval duration.

## Principal component analysis

We found three principal components corresponding to 71% of the total variance. The first one, corresponding to 31.4% of variance, reflected global improvement in ventricular function and natriuretic biomarker reduction, being significantly higher in the treatment group, indicating greater overall benefit with pentoxifylline. The second one, corresponding to 22.4% of variance, characterizes an inflammatory–perfusion axis. The last one, corresponding to 17.4% of variance, captured the interplay between changes in inflammation and contractility. No significant difference in these components were observed between groups (p > 0.05).

## Discussion

The main findings of this study testing the effect of use of PTX in patients with CCC showed a trend toward reducing serum levels of pro-inflammatory cytokines, such as IL-6 and TNF-α, and increasing serum levels of the anti-inflammatory cytokine IL-10. Despite these positive effects suggesting a potential benefit in modulating the inflammatory profile in CCC patients, we did not observe significant effects of PTX on myocardial perfusion, cardiac function, or structure. The component representing improvement in systolic function, quality of life, and natriuretic peptide reduction was the only axis to distinguish the pentoxifylline and placebo groups, supporting a beneficial effect of pentoxifylline on global cardiac performance in chronic Chagas patients. The absence of group differences in the other principal components suggests that modulation of inflammatory markers, perfusion defects, and contractility was not detectable by principal component analysis, maybe due to limited sample size. These findings highlight that pentoxifylline's primary impact may lie in improving ventricular mechanics and patient-reported outcomes.

### Study population characteristics

The demographic characteristics of our cohort were similar to those of typical Chagas populations, with a marked predominance of male patients. The mean age was over 50 years, which aligns with the populations with CCC reported in other studies, likely due to the long period between infection and the development of cardiomyopathy [32].

Regarding LVEF, the study population showed only a mild reduction in global left ventricular systolic function at baseline, as expected based on the study's inclusion criteria. This criterion was implemented to exclude patients with more advanced myocardial dysfunction, in which secondary injury mechanisms, such as intense reflex neurohormonal activation in heart failure, could result in additional inflammatory alterations and myocardial damage, potentially confounding the assessment of inflammatory mechanisms more closely associated with the inherent pathophysiology of CCC. Thus, the majority of the patients had mild cardiomyopathy, in early or intermediate stages of CCC, corresponding to stages B1, B2, and C of CCC progression [33]. Patients were mostly asymptomatic or had mild symptoms, with 58.7% classified as NYHA Functional Class I at the time of enrollment.

### Effects of pentoxifylline on serum cytokines

As pointed out above, the serum cytokine levels presented in Table 2, demonstrate that patients treated with PTX showed a strong trend toward an increase in IL-10, an anti-inflammatory cytokine, suggesting a favorable effect of the drug in modulating the inflammatory mechanisms present in the chronic phase of the disease. Additionally, we observed a trend toward a reduction in TNF-α levels in the PTX-treated group (p = 0.06), further supporting the anti-inflammatory effect of

the drug in CCC, as seen in other studies [13]. In our study, pentoxifylline had a neutral effect on all other cytokines evaluated (IL-6, endothelin-1, IFN-γ, and TGF-β).

A meta-analysis evaluating the effects of pentoxifylline on pro-inflammatory cytokines in patients with dilated cardiomyopathy of various causes (ischemic, idiopathic, and hypertensive) showed a significant reduction in plasma TNF-α concentrations and no change in IL-6 levels [34]. The effect of PTX on TNF-α in dilated cardiomyopathies is controversial, with studies in the literature showing no effect of the drug on serum levels of this marker [25,35].

It is plausible that the trends observed in our study, showing a reduction in TNF-α and an increase in IL-10 without reaching statistical significance, may be due to the limited sample size. However, it is also important to consider that these trends may have occurred by chance, in line with the small population sample size.

### Effects of pentoxifylline on NT-ProBNP

No significant effects of pentoxifylline on NT-ProBNP levels were observed. Values remained similar between the placebo and treatment groups, with no statistically significant changes detected in the measurements either before or after treatment.

This biomarker, which is secreted by the myocardium in response to myocardial fiber stretch, reflects diastolic overload of the ventricular chambers and has been recognized as a prognostic marker in cardiomyopathy of various etiologies [36].

Several recent studies have evaluated NT-proBNP levels in patients with CCC, highlighting its role as a prognostic biomarker for cardiac damage and disease progression. A key study found that elevated NT-proBNP levels were significantly associated with adverse cardiovascular outcomes and mortality in patients with CCC, underscoring its value as a predictor of disease severity [37]. This finding was confirmed in a cohort study, where NT-proBNP was one of the biomarkers most strongly associated with the risk of cardiovascular events, including death, in CCC patients [38]. Additionally, another study indicated that higher NT-proBNP levels were associated with worse clinical outcomes in asymptomatic patients with Chagas disease, suggesting its utility even in the early stages of the disease for predicting future cardiac complications [39].

The neutral results regarding NT-proBNP values found in our study are consistent with other results of the present study that indicated a lack of beneficial effect of pentoxifylline on left and right ventricular systolic function, ventricular diameters, and left ventricular diastolic function.

### Effects of pentoxifylline on cardiac structure and function

At the baseline assessment, we observed a predominance of wall motion abnormalities in the inferior and inferolateral segments, as well as the apical segments. This preferential area of involvement is a characteristic feature of CCC. We did not observe the presence of left ventricular (LV) apical aneurysms in any of the studied patients, which can be explained by the predominance of mild cardiomyopathy in this sample. Different series of patients with CCC, investigated with echocardiography, have shown that the average prevalence of apical aneurysms in patients with mild cardiomyopathy was 8.5% (ranging from 1.6% to 8.6%), in contrast to up to 55% in patients with moderate to severe LV systolic dysfunction [40].

In the post-treatment assessment, our results, presented in Table 4, demonstrated a neutral effect of PTX on cardiac structure and function. Reviewing the literature, we observed that the results of PTX's effects in clinical and experimental settings are quite heterogeneous.

A study by Tanaka et al., which tested the use of PTX in rodents, did not demonstrate improvement in ventricular function. The study was conducted in Syrian hamsters divided into three groups: two groups infected with *T. cruzi* (6 months post-experimental infection), one of which was treated with PTX and the other with saline, as well as a non-infected control group. The animals underwent an initial evaluation with echocardiography and myocardial perfusion scintigraphy and were reassessed with the same methods after 60 days. At the end of the study, the PTX group presented perfusion defect

areas similar to those observed in the control group. However, the attenuation of perfusion defect progression did not prevent the progression of left ventricular dysfunction [41]. Another important study conducted with hamsters chronically infected by *T. cruzi* did not find beneficial effects on ventricular function with the use of etanercept, another TNF-α synthesis blocker [42].

In contrast, a study using mice infected with the Colombian strain of *T. cruzi* showed restoration of LV ejection fraction (LVEF) after 30 days of treatment with PTX. In this study, PTX improved cardiac remodeling by reducing left ventricular hypertrophy and restoring LVEF. It also improved critical aspects of CCC and shifted the CD8+T-cell response towards homeostasis, reinforcing that immunological abnormalities are crucially linked, whether as cause or consequence, to CCC [20].

In the clinical context, studies investigating the effects of pentoxifylline on ventricular function are scarce and have also shown conflicting results.

An important study conducted with patients with idiopathic dilated cardiomyopathy showed an improvement in functional class and LVEF after 6 months of treatment with PTX compared to the placebo group (mean LVEF 38.7% [SD 15.0] vs. 26.8% [SD 11.0], p=0.04) [24]. Another study by the same group, involving 38 patients with ischemic cardiomyopathy, showed that PTX significantly improved LVEF and reduced inflammatory markers such as TNF-α and C-reactive protein (CRP) [43].

Conversely, a study with 47 patients with heart failure of various causes (31.9% ischemic, 21.3% hypertensive, 10.6% ischemic and hypertensive, 36.2% idiopathic dilated cardiomyopathy) randomly assigned patients to pentoxifylline 600mg BID (n=23) or placebo (n=24). All patients had compensated heart failure with LVEF ≤ 40% and had to have been receiving therapy for at least 3 months. After 6 months of treatment, LVEF did not change in the pentoxifylline group compared to placebo (29±7% to 33±10% vs. 27±9% to 34±9%, respectively) [35].

These studies suggest that pentoxifylline may have beneficial effects in some forms of heart failure, particularly by reducing inflammation and improving ventricular function. However, the results vary depending on the etiology of cardiomyopathy and the severity of the disease.

One hypothesis to explain our neutral results regarding PTX's effects on cardiac structure and function is that the duration of treatment may have been insufficient to observe benefits in these variables. In fact, CCC is characterized by a chronic low-grade inflammatory process, with myocardial damage that develops and progresses slowly over time. A cohort of CCC patients without advanced cardiomyopathy, based on echocardiographic assessments, suggested that it would take about 5 years to detect a significant reduction in LVEF in these patients [31]. Therefore, it is possible that a longer treatment period, perhaps several years, would show a positive impact of this medication on cardiac function and structure. However, a long-term study would face inherent challenges, such as costs and reduced patient adherence over time.

Another factor to consider in explaining our results is the mild degree of ventricular dysfunction in our patient cohort. It is still unclear whether a cohort with more advanced cardiomyopathy, and therefore greater myocardial fibrosis and structural damage, would show a greater impact from an immunomodulatory drug like PTX. Therefore, studies involving patients with more severe CCC should be conducted to compare the effects of PTX at different stages of CCC progression.

### Effects of pentoxifylline on myocardial perfusion

No significant differences were observed when comparing the areas of fixed perfusion defects at rest and under stress between patients in either of the study groups. Although we observed a reduction in the areas of reversible defects in the PTX group, this difference did not reach statistical significance, hence preventing us from concluding that pentoxifylline had a favorable effect on reversing ischemic perfusion defects in our study.

In the experimental field, Tanaka et al. tested the effects of prolonged PTX use on myocardial perfusion in Syrian hamsters and found that PTX attenuated the progression of perfusion defects in animals chronically infected with *T. cruzi*, while the group of infected animals treated with saline showed progression of the perfusion defect areas [41].

This apparent discrepancy between the results of preclinical studies, obtained in the Syrian hamster model of CCC, and the present findings could be explained by the more pronounced inflammation in the experimental model compared to human CCC. On the other hand, intrinsic differences in the effects of PTX between rodents and humans could also contribute to the discrepant observed results.

## Effects of pentoxifylline on the electrocardiogram

Administration of pentoxifylline during this experimental protocol resulted in neutral effects on the electrocardiographic variables analyzed. The prevalence of complete right bundle branch block observed in this cohort was like that described in previous studies in Chagas disease populations [44], with no significant changes after treatment in either group. As expected, intraventricular conduction disturbances, particularly right bundle branch block and left anterior fascicular block, remained the most frequent electrocardiographic alterations. A low prevalence of atrial fibrillation (AF) was observed in both groups, in line with the literature indicating that AF is uncommon in chronic Chagas cardiomyopathy and usually associated with heart failure and a worse prognosis. Large cohort studies have reported rates between 2.5% and 3.1% of AF over long-term follow-up [45,46]. No supraventricular ectopic beats were identified in any patient, possibly due to the mild stage of cardiomyopathy in this sample, as atrial arrhythmias tend to occur in more advanced disease. Ventricular ectopic beats showed a reduction at the final assessment in both groups, though this did not reach statistical significance. Quantitative electrocardiographic variables also showed no effect of pentoxifylline on corrected QT interval, QT dispersion, QRS duration, or PR interval. These findings contrast with experimental studies in animal models that demonstrated beneficial effects of pentoxifylline on conduction parameters [20]. Long-term follow-up studies have indicated that QT interval parameters are potential prognostic markers of arrhythmic risk and mortality in chronic Chagas disease [47,48]. In this study, patients in both groups exhibited increased QT dispersion at baseline, likely reflecting myocardial fibrosis and electrical heterogeneity [49]. Pentoxifylline therapy did not reduce QT dispersion or corrected QT duration, suggesting that its immunomodulatory action was insufficient to modify ventricular repolarization as assessed by these electrocardiographic markers

## Effects of pentoxifylline on health-related quality of life

The assessment of health-related quality of life in our cohort shows that patients in both study groups had high mean scores (greater than or equal to 70) in most domains (functional capacity, physical aspects, pain, social aspects, emotional aspects, and mental health), indicating a mild impact of CCC on quality of life. This is consistent with the non-advanced cardiomyopathy present in the patients studied. The domains most affected in both groups of CCC patients were general health and vitality.

Interestingly, in our study, we observed a significant improvement in the functional capacity of patients treated with PTX. This may be attributed to the extracardiac effects of pentoxifylline, such as its impact on muscle and peripheral vasodilation. A study conducted on rats with heart failure tested the effects of PTX on skeletal muscle blood flow and vascular conductance at rest and during submaximal exercise. That study showed significant improvements in these parameters during locomotor exercise [50].

There are numerous reports on the effects of PTX in improving the rheological properties of blood. This effect is related to several factors, including reduced plasma and whole blood viscosity (largely due to decreased plasma fibrinogen), increased erythrocyte deformability, decreased erythrocyte and platelet aggregation, and improved blood filterability. The cumulative effect of these factors leads to enhanced capillary blood flow [51,52]. Pentoxifylline also exhibits anti-inflammatory and antioxidant effects. Its antioxidant effects seem to be primarily due to reduced neutrophil activation, as activated neutrophils generate superoxide via NADPH oxidase [53]. The combination of these effects may help to explain the improvement in functional capacity observed in the patients in our study.

## Conclusions

The results of this pilot study suggest a potential positive effect of PTX in modulating the inflammatory profile of patients with CCC.

However, the use of pentoxifylline over 6 months in patients with CCC was not sufficient to attenuate the degree of ventricular dysfunction or reduce myocardial perfusion defects.

Additionally, results of principal component analysis suggest that pentoxifylline might improve global cardiac function and quality of life, warranting further investigation in larger cohorts to clarify its impact on inflammatory and perfusion parameters.

Given these findings, a clinical trial with a longer treatment duration and larger sample size is warranted to provide a more accurate assessment of PTX's effects in CCC patients.

## Study limitations

The study sample size was smaller than initially planned, and this could have caused a reduction in the power of the statistical analysis, mainly in the cytokines results. The follow up period of this study was relatively short, considering the natural history of the CCC, and could be responsible for the neutral results of PTX on cardiac structure and function.

## Supporting information

**S1 Checklist. CONSORT 2010 checklist of information to include when reporting a randomized trial.** This checklist is adapted from the CONSORT 2010 checklist, which is licensed under a Creative Commons Attribution 4.0 International License (CC BY 4.0; http://creativecommons.org/licenses/by/4.0/).
(S1_Checklist.DOC)

## Author contributions

**Conceptualization:** Káryta Suely Macêdo Martins.

**Data curation:** Káryta Suely Macêdo Martins, Denise Mayumi Tanaka, Henrique Turin Moreira, Marcus Vinícius Simões.

**Formal analysis:** Káryta Suely Macêdo Martins, Denise Mayumi Tanaka, Henrique Turin Moreira.

**Funding acquisition:** José A. Marin-Neto.

**Investigation:** Káryta Suely Macêdo Martins, Denise Mayumi Tanaka, Camila Godoy Fabricio, Antônio Carlos Leite de Barros Filho, Henrique Turin Moreira, Paulo Louzada Júnior, José A. Marin-Neto.

**Methodology:** Káryta Suely Macêdo Martins, Marcus Vinícius Simões.

**Project administration:** Káryta Suely Macêdo Martins, José A. Marin-Neto, Marcus Vinícius Simões.

**Resources:** Denise Mayumi Tanaka.

**Supervision:** Marcus Vinícius Simões.

**Visualization:** Denise Mayumi Tanaka.

**Writing – original draft:** Káryta Suely Macêdo Martins, Marcus Vinícius Simões.

**Writing – review & editing:** Káryta Suely Macêdo Martins, Denise Mayumi Tanaka, Camila Godoy Fabricio, Antônio Carlos Leite de Barros Filho, Henrique Turin Moreira, Paulo Louzada Júnior, José A. Marin-Neto, Marcus Vinícius Simões.

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
