## [Decision Letter · Decision Letter 0]

4 Feb 2025

PNTD-D-24-01327

Effects of prolonged pentoxifylline in patients with chronic Chagas cardiomyopathy: a randomized, double-blind, controlled pilot trial.

Dear Dr. Simoes,

Thank you for submitting your manuscript to PLOS Neglected Tropical Diseases. After careful consideration, we feel that it has merit but does not fully meet PLOS Neglected Tropical Diseases's publication criteria as it currently stands. Therefore, we invite you to submit a revised version of the manuscript that addresses the points raised during the review process.

Please submit your revised manuscript within 60 days Apr 05 2025 11:59PM. If you will need more time than this to complete your revisions, please reply to this message or contact the journal office at plosntds@plos.org. Please include the following items when submitting your revised manuscript:

We look forward to receiving your revised manuscript.

Kind regards,

Norman L. Beatty, MD

Academic Editor

Claudia Brodskyn

Section Editor

Shaden Kamhawi

co-Editor-in-Chief

Paul Brindley

co-Editor-in-Chief

**Additional Editor Comments:**

Thank you for submitting your manuscript for consideration to publish in PLOS NTD. The research conducted is important to the field of clinical Chagas disease and for those individuals impacted by this devastating infectious process. The authors have a designed a well thought out clinical trial utilizing pentoxifylline in those with chronic Chagas cardiomyopathy. The findings show promise that pentoxifylline may be an agent that could be utilized to improve the life of those living with chronic Chagas disease and also possibly alter the pathophysiology of this process. However, as it currently stands the findings needs additional analysis and improved presentation prior to being considered for publication. It is very important that the authors openly discuss the limitations of this study and address the small sample size. Reviewer 3 recommends a principal component analysis (PCA) of each group. This is a reasonable request, and I believe it will improve the presentation of your results.

Additional academic editor comments:

- Title needs to be revised. I agree with Reviewer 3 that the use of "prolonged" is not precise. I do agree that keeping "pilot trial" in the title is important.

- Please add limitations to the conclusions.

- Please provide a detailed author response document which addresses the suggestions and comments from each Reviewer.

**Journal Requirements:**

At this stage, the following Authors/Authors require contributions: Karyta Suely Macedo Martins, Denise Mayumi Tanaka, Camila Godoy Frabricio, Antonio Carlos Leite de Barros Filho, Henrique Turin Moreira, Paulo Louzada Jr, Jose Antonio Marin-Neto, and Marcus Vinicius Simoes. Please ensure that the full contributions of each author are acknowledged in the "Add/Edit/Remove Authors" section of our submission form.

- ® on Lines: 179, and 234.

5) Please ensure that all Figure files have corresponding citations and legends within the manuscript. Currently, Figures 4 and 5 in your submission file inventory do not have an in-text citation.

6) Please ensure that all Table files have corresponding citations and legends within the manuscript. Currently, Table 6 in your submission file inventory does not have an in-text citation.

7) We have noticed that you have uploaded Supporting Information files, but you have not included a list of legends. Please add a full list of legends for your Supporting Information files after the references list.

8) We note that your Data Availability Statement is currently as follows: "Yes, the data will be available.". Please confirm at this time whether or not your submission contains all raw data required to replicate the results of your study. Authors must share the “minimal data set” for their submission. PLOS defines the minimal data set to consist of the data required to replicate all study findings reported in the article, as well as related metadata and methods (https://journals.plos.org/plosone/s/data-availability#loc-minimal-data-set-definition).

- The points extracted from images for analysis..

9) Please ensure that the funders and grant numbers match between the Financial Disclosure field and the Funding Information tab in your submission form. Note that the funders must be provided in the same order in both places as well.

**Reviewers' Comments:**

Reviewer's Responses to Questions

**Key Review Criteria Required for Acceptance?**

**Methods** :

-Are the objectives of the study clearly articulated with a clear testable hypothesis stated?

-Is the study design appropriate to address the stated objectives?

-Is the population clearly described and appropriate for the hypothesis being tested?

-Is the sample size sufficient to ensure adequate power to address the hypothesis being tested?

-Were correct statistical analysis used to support conclusions?

-Are there concerns about ethical or regulatory requirements being met?

Reviewer #1: Are the objectives of the study clearly articulated with a clear testable hypothesis stated? Yes

Is the study design appropriate to address the stated objectives? Yes

Is the population clearly described and appropriate for the hypothesis being tested? Yes

Is the sample size sufficient to ensure adequate power to address the hypothesis being tested? No, I suggest the authors to include the limitations and repercussion of not including the number of sample size planned.

Were correct statistical analysis used to support conclusions? Yes

Are there concerns about ethical or regulatory requirements being met? Yes

Reviewer #2: - Methods are well-described and detailed.

- Study design (randomized-controlled trial) is appropriate and statistical analyses seem correctly done.

- The main weakness of the paper is the small sample size. Appropriate power calculation were performed but the assumptions used were likely too small. a 5% difference in LVEF is likely within the test-to-test variability of echocardiography. In fact, it is actually smaller than the 6% variability the authors provide, which also seems smaller than I would expect and is not supported by a citation.

- The randomization scheme needs to be described in more detail.

- Consider reporting interquartile range instead of total range.

Reviewer #3: Yes

**Results** :

-Does the analysis presented match the analysis plan?

-Are the results clearly and completely presented?

-Are the figures (Tables, Images) of sufficient quality for clarity?

Reviewer #1: Does the analysis presented match the analysis plan? Yes

Are the results clearly and completely presented? I suggest the authors to compare the baseline NT-ProBNP level between placebo and PTX group because they are quite diferents and comment this difference

Are the figures (Tables, Images) of sufficient quality for clarity? Yes

Reviewer #2: - Overall the data is presented clearly. A CONSORT diagram is included.

- Participants who died were excluded from analyses, but cause of death should be provided (if available) and discussed.

- While it is unlikely that they affected the outcome, especially since there were similar numbers in each group, including lost to follow up and dead participants in a sensitivity analysis would strengthen the paper. This is especially important given that the overall sample size was so small.

- Prevalence of hypertension and diabetes were much higher in the placebo group than in the control group, which likely biased results since it does not appear that these comorbidities were adjusted for when comparing outcomes. Even though this was an RCT, these differences probably should be adjusted for as they almost certainly affect LVEF trajectories.

- The authors do not describe how they defined systemic hypertension. Was this self reported? the mean BP for each group was in the normal range despite a hypertension prevalence of 89% in the placebo group.

Reviewer #3: Yes. See comments

**Conclusions** :

-Are the conclusions supported by the data presented?

-Are the limitations of analysis clearly described?

-Do the authors discuss how these data can be helpful to advance our understanding of the topic under study?

-Is public health relevance addressed?

Reviewer #1: Are the conclusions supported by the data presented? Yes

Are the limitations of analysis clearly described? No, I suggest to include this topic.

Do the authors discuss how these data can be helpful to advance our understanding of the topic under study? Yes

Is public health relevance addressed? Yes

Reviewer #2: - The observed changes in cytokine levels are not surprising since pentoxifylline is an anti-inflammatory. In this study, it is not really possible to connect these changes with any clinical outcomes.

- They admit that the short duration of follow up (6 months) may have been too short to detect a clinically significant decline in EF, which is more likely to occur on a scale of years.

Reviewer #3: Yes. Limitation section is required. See comments

**Editorial and Data Presentation Modifications?**

Reviewer #1: Please include references in the last paragraph of the topic - Effects of Pentoxifylline on NT-ProBNP

Reviewer #2: - Figure 2 could be removed.

- line 412: I think they are referring to table 5 not table 4.

- Figure 4 and 5 are not really necessary and could be deleted or moved to a supplement.

- line 483: I think they are referring to table 6 not table 5.

Reviewer #3: See comments

**Summary and General Comments** :

Reviewer #1: This study complements the research of pentoxifilin treatment specific for CCC patients, therefore it is relevant.

Reviewer #2: This study was a well-designed and conducted randomized clinical trial of the effect of pentoxifylline on LVEF in patients with Chagas cardiomyopathy. The methods are generally sound though there are some areas that need clarification or more in-depth description, such as the randomization scheme. The statistical analyses are appropriate but might be easier to understand with a little more explanation of the mixed-effects ANOVA models. Additionally, since the two groups were unbalanced with respect to multiple potential confounders, these covariates should probably be included in models. Unfortunately, the small sample size in the study is a significant limitation. The effect size they used to calculate sample size is quite small and may not be clinically relevant (or within the variation of the test). The combination of the very small sample size and imbalance in several key confounders at baseline make it nearly impossible to draw conclusions from the results.

Reviewer #3: General comments:

I would like to thank the editors and authors for the opportunity to contribute by reading and commenting on this MS. The MS is very interesting, and I have tried to enhance my comments to the level of this contribution.

Chagas disease is a neglected disease and several aspects of its pathogenesis, diagnosis and therapy deserve much attention. In this parasitic disease, the time for etiological therapy is often lost due to the alleviated acute phase. The patient's fate is unclear and in the chronic phase the therapeutic regimen used only attenuates the clinical signs and symptoms. There is an urgent need for rational therapies, particularly considering the pathogenic mechanisms proposed to underly the most frequent form of Chagas disease, chronic cardiomyopathy. Based on the current corollary that pathogenesis of the cardiac form of Chagas disease is underlined by (i) the presence of scarce parasitism, (ii) low-grade local inflammation, (iii) systemic inflammatory profile and (iv) intrinsic cardiac tissue injuries, the proposal of a therapeutic approach focused on regulating the unbalanced immune response deserves much attention and careful interpretation of the results.

The here presented therapeutic proposal for CCC using pentoxifylline is based on a hypothesis supported by previous findings in non-infectious human cardiomyopathies and preclinical studies using mouse and hamster models of chronic Chagas cardiomyopathy. In the present study, the therapeutic regimen is the commonly used (pentoxifylline - 400 mg 3x/day), but the therapy duration (“prolonged”, “long-term”) is a point to be discussed (was it sufficient?). The parameters analyzed are very well chosen, the experimental design is adequate (baseline versus endpoint; proposed therapy versus placebo, double blinded analysis). However, the results are not well shown, and some information may be lost (showing data only in means +/- SD). Therefore, the MS cannot be accepted in its current format and requires major revision.

Lastly, improving quality of life is a very relevant issue for Chagas disease patients. I am glad it was considered in the present study.

Specific comments:

Title:

Line 1 - “Effects of prolonged…”. “Prolonged” is an imprecise adjective and should be avoided in the title. Particularly considering that 6 months of PTX administration regimen was used for non-infectious ischemic cardiomyopathy (Sliwa et al 2004) and it was not considered a long-term use. Further, the chronic Chagas cardiomyopathy flourishes after decades of parasitism and inflammatory lesions. Due to the nature of the infection and the individual differences, this reviewer should expect the need for a longer therapeutic regimen to observe more prominent results in Chagas disease. Please, consider removing this word of the title. Further, a support for the use of this therapy duration should be bring in the introduction section and this point should be considered in Discussion section.

Abstract:

Data presentation (means +/- SD) may put down relevant findings (particularly considering “individuals”) and may indicate that the sample size calculation was inappropriate. Unfortunately, this appears to be the case (see comments: Results).

Line 57: “Long-term” (here used as synonyms of prolonged) – should be considered for exclusion from this sentence.

Introduction:

This section is mostly well-written.

Line 73: “The chronic phase, which is symptomatic…” most patients with Chagas disease are asymptomatic. Even among patients with cardiac disease, many are asymptomatic, although EKG and/or ECHO abnormalities may be detected when patients are clinically evaluated. The entire sentence (lines 73–76) should be rephrased to avoid misleading readers.

Lines 82-86: This sentence brings a restrict and doubtful point of view. Please, rephase it and, at least, present a divergent point of view.

Kierszenbaum in his article “Where do we stand on the autoimmunity hypothesis of Chagas disease? (Trends Parasitol. 2005 Nov;21(11):513-6. doi: 10.1016/j.pt.2005.08.013) assumed that this deleterious idea (here presented as “probably autoimmunity mechanisms” supported by a review of the group) has delayed research in Chagas disease. “The notion that autoimmunity underlies pathogenesis in Chagas disease has had a negative impact on efforts to develop effective chemotherapy and vaccines against Trypanosoma cruzi infection.” I invite the authors to read Kierszenbaum's article and to revisit the literature in depth in search of support for autoimmunity in Chagas disease.

The proposed use of PTX in Chagas disease may find support in data showing chronic cardiac injury enriched in TNF+ cells (Higuchi’s studies) and in the dysregulated “systemic inflammatory profile” found in sera of patients with heart disease (Nunes et al., 2022 and earlier by Ferreira et al., Perez-Fuentes et al., Perez et al., Dutra et al.), which is supported by findings in preclinical models showing high levels of inflammatory cytokines in plasma and cardiac tissue.

The choosing of cytokines as inflammatory mediators used as readouts for systemic inflammation should be better supported.

Line 120: Again, “the prolonged” should be supported. Why do the authors consider 6 months a long-term treatment?

Methods and Population.

Electrocardiograms are routine and the data should be presented as they can provide relevant information.

Lines 225-159: It is not my field of study, and the images and information are very good and help to follow the results. However, if standardized protocols were used for myocardial perfusion analysis (cited reference #28), why were Fig 1 and Fig 2 shown in Methods? Is there a need to have any specific analysis/interpretation for Chagas disease cardiopathy? If yes, please use it as the result. If not, please use it as supporting information.

Line 271: I could not find “Appendix 1”. If the questionnaire is available online in English version, please indicate link for access.

The study population was classified as NYHA I or II. Have you included patients with severe CCC (like C patients, according to the Guidelines by Marin-Neto et al, 2023). Why was LVEF the parameter (5% difference in 6 months?) used to calculate sample size? Please, explain it in the text and support it by reference.

Results

EKG are routine and the data should be presented as they can provide relevant information.

Table 1: Was the disequilibrium of male gender, hypertension, and NYHA Class II between the PLC versus PTX groups corrected?

Table 1. Patients are classified as NYHA I or II are using ACE inhibitors, β-blockers, diuretics, spironolactone. Continued throughout the study period? Any other medication? Hormones (particularly considering gender imbalance – and the effects of hormones on cytokine expression/levels)? Considering the parasitological origin of the disease, were they treated etiologically (benznidazole? other?)? Recent data support that prior treatment with antiparasitic medication (benznidazole) may influence TNF levels.

Table 2: The results of cytokine evaluation are interesting. The small number of patients per group may contribute to explaining p-values, and the analysis used (data are only shown as means+/- SD). Considering the common variations of cytokine levels in serum of Chagas disease patients (a quite individual disease that may be influenced by parasite diversity, period after infection, neuro-hormonal and immunological conditions of patients), data should be presented as individuals using graphs showing levels before and after treatment, at least for TNF, IL-6 and IL-10.

The endothelin-1 levels should be reviewed (it seems quite irreal = ED-1 (pg/mL) 4.37 ± 98 // 5.58 ± 15.2). Are they correct?

Lazzerini et al proposal of the contribution of heart inflammation in prolonged QTc syndrome, and the participation of EKG alterations found in Chagas disease patients, particularly prognostic value of QT interval (Salles et al. 2003), were the EKG analyzed in these groups of patients? Has PTX any impact on EKG parameters, particularly on QT/QTc interval?

Table 3: At baseline, pro-BNP levels are very dissimilar. PLC = 1207.1 ± 2343.5 versus PTX = 654.9 ± 1226.9. The final assessment also shows differences PLC = 2762.8 366 ± 8098.4 versus PTX = 508.4 ± 464.7. What should these differences be due to? Were all dosages performed at the same time and using the same kit/plate/reagents?

Line 377: do you mean Table 4? Line 412: do you mean Table 5?

Table 5: The importance of r myocardial perfusion deficits (defects?) justifies the presentation of data as individuals (graphs before and after treatment). I could not find the results of ischemic deficits before and after treatments.

Table 6. CF = functional capacity, CF should be FC; AF = physical aspects, AF should be PA / PhA

Considering the number of patients, the borderline p-values seen in some parameters analyzed, the biological aspects (when the gain of 5-10% of beneficial response in a 6-month interval may have importance for the patient's quality of life), I suggest that the authors take a multiparameter view of each group of patients in a principal component analysis (PCA). This analysis can add a more understandable view of the results. Furthermore, it can establish any possible correlation between variables.

Considering the high relevance of this study and the 5-year period after patient inclusion/therapy, is it possible to bring survival surveillance and/or record cardiovascular events of this small group of patients? This could provide relevant information.

Discussion

This section is mostly well-written and may gain with the revision of Results.

Lines 460-462 versus Line 494: I agree that IL-6 is an interesting cytokine to be evaluated and considered in the pathogenesis of CCC (particularly considering the type of analysis and graphs and, moreover, the multiparametric analysis I have suggested). However, these 2 sentences are not coherent. Please, revise it after the suggested analysis.

Line 492: It is not coherent. Table 2 shows p = 0.06 (“…TNF-α levels in the PTX-treated group (p=0.05)”). If you used different models of analysis, you should present them.

Line: 481-482: “Thus, the majority of the patients had mild cardiomyopathy, in early or intermediate stages of CCC, corresponding to stages B1, B2 and C of CCC progression”. However, according to The SBC Guideline on the Diagnosis and Treatment of Patients with Cardiomyopathy of Chagas Disease – 2023 (Marin-Neto et al. 2023), the patients classified as C patients have severe form of CCC (Table 5.2: Structural cardiac disease; global systolic ventricular dysfunction; previous or current symptoms of HF), although they may be classified as NYHA class I, II, III or IV. This makes me wonder: 1- Are you comparing comparable groups of patients? (Table 1 shows more NYHA class I (maybe B1/B2?) in the PTX group); What is the percentage of C patients in each group? 2- Why did you adopt the NYHA classification if Brazil has a First (2011) and a Second (2023) Guidelines for CCC classification?

Lines 549-568: These are very interesting points. I am glad that you discussed preclinical models, which are often ignored in MS focused on clinical trials or clinical CCC. Preclinical models reproduce aspects of CCC (particularly those that are highly conserved across species). I appreciate the comment that Etanercept is not a good TNF blocker. Another preclinical study using a better TNF blocker (Infliximab) found very interesting results in a CCC model.

However, the duration of treatments (30 or 60 days in models) should not be considered to compare with the 6-month treatment used in the present study. The progression of CCC in models and humans is not comparable. In humans, CCC is a long-lasting and very heterogeneous disease, in contrast to the preclinical model. Even compared to any other human cardiomyopathies (ischemic, for example), CCC is a long-lasting process (probably with different pathogenic mechanisms, as parasite persistence) and is expected to require longer therapy, as you have considered in Discussion.

A Limitation section is required.

PLOS authors have the option to publish the peer review history of their article (what does this mean? ). If published, this will include your full peer review and any attached files.

**Do you want your identity to be public for this peer review?** For information about this choice, including consent withdrawal, please see our Privacy Policy .

Reviewer #1: No

Reviewer #2: No

Reviewer #3: No

**Figure resubmission:**

**Reproducibility:**



---

## [Editor Report · Decision Letter 1]

11 Sep 2025

Dear Dr. Simoes,

We are pleased to inform you that your manuscript 'Effects of pentoxifylline in patients with chronic Chagas cardiomyopathy: a randomized, double-blind, controlled pilot trial.' has been provisionally accepted for publication in PLOS Neglected Tropical Diseases.

Best regards,

Norman Beatty, MD

Academic Editor

Claudia Brodskyn

Section Editor

Shaden Kamhawi

co-Editor-in-Chief

Paul Brindley

co-Editor-in-Chief

Thank you for providing a detailed author response document and revised manuscript. The three peer reviewers and academic editor provided suggestions and comments for which the author team has sufficiently revised their manuscript accordingly. Limitations of the study are discussed, and title was adjusted based off recommendations. Additional material added and some errors were corrected. Results of this pilot study could support evidence for a larger scale investigation of PTX among those with chronic Chagas disease cardiomyopathy. I recommend acceptance at this time.

---

## [Editor Report · Acceptance letter]

Dear Dr. Simoes,

We are delighted to inform you that your manuscript, " 

Effects of pentoxifylline in patients with chronic Chagas cardiomyopathy: a randomized, double-blind, controlled pilot trial.," has been formally accepted for publication in PLOS Neglected Tropical Diseases.

Best regards,

Shaden Kamhawi

co-Editor-in-Chief

Paul Brindley

co-Editor-in-Chief
